# The Effects of Silver-Releasing Foam Dressings on Diabetic Foot Ulcer Healing

**DOI:** 10.3390/jcm10071495

**Published:** 2021-04-03

**Authors:** Yu-Chi Wang, Hsiao-Chen Lee, Chien-Lin Chen, Ming-Chun Kuo, Savitha Ramachandran, Rong-Fu Chen, Yur-Ren Kuo

**Affiliations:** 1Division of Plastic and Reconstructive Surgery, Department of Surgery, Kaohsiung Medical University Hospital, Kaohsiung 80756, Taiwan; heidi.ycw@gmail.com (Y.-C.W.); s640125@yahoo.com.tw (H.-C.L.); chienlinch@gmail.com (C.-L.C.); savitha.rama@gmail.com (S.R.); dr.chenrf@gmail.com (R.-F.C.); 2Division of Endocrinology and Metabolism, Department of Internal Medicine, Kaohsiung Chang Gung Memorial Hospital, Kaohsiung 83301, Taiwan; kmc4047@adm.cgmh.org.tw; 3Department of Plastic and Reconstructive Surgery, KK Women’s and Children’s Hospital, Singapore 229899, Singapore; 4Faculty of Medicine, College of Medicine, Kaohsiung Medical University, Kaohsiung 80708, Taiwan; 5Department of Biological Sciences, National Sun Yat-Sen University, Kaohsiung 80424, Taiwan; 6SingHealth Duke-NUS Musculoskeletal Sciences Academic Clinical Programme, Singapore 168753, Singapore

**Keywords:** diabetic foot ulcer, silver-releasing foam, silver sulfadiazine, wound healing

## Abstract

Diabetic foot ulcers (DFUs) are a serious complication in diabetic patients and lead to high morbidity and mortality. Numerous dressings have been developed to facilitate wound healing of DFUs. This study investigated the wound healing efficacy of silver-releasing foam dressings versus silver-containing cream in managing outpatients with DFUs. Sixty patients with Wagner Grade 1 to 2 DFUs were recruited. The treatment group received silver-releasing foam dressing (Biatain^®^ Ag Non-Adhesive Foam dressing; Coloplast, Humlebaek, Denmark). The control group received 1% silver sulfadiazine (SSD) cream. The ulcer area in the silver foam group was significantly reduced compared with that in the SSD group after four weeks of treatment (silver foam group: 76.43 ± 7.41%, SSD group: 27.00 ± 4.95%, *p* < 0.001). The weekly wound healing rate in the silver foam group was superior to the SSD group during the first three weeks of treatment (*p* < 0.05). The silver-releasing foam dressing is more effective than SSD in promoting wound healing of DFUs. The effect is more pronounced in the initial three weeks of the treatment. Thus, silver-releasing foam could be an effective wound dressing for DFUs, mainly in the early period of wound management.

## 1. Introduction

Diabetes mellitus has been a growing public health issue worldwide due to significant risk for mortality and morbidity because of diabetic foot ulcers (DFUs) and associated lower extremity problems. The global prevalence of DFU is 6.3% [1]. Among diabetic patients, the annual incidence of foot ulcers is 1% to 4% and the lifetime prevalence of foot ulcers is 15% to 25% [1]. More than half of DFUs eventually become infected, which contributes to 84% of diabetic limb amputations and has a 39% to 80% risk of mortality 5 years after amputation [2,3]. This causes poor quality of life for patients and leads to a large socioeconomic burden. Therefore, early management of DFUs is critical to prevent ulcer exacerbation and subsequent complications.

Pathophysiology of DFU is multifactorial [4,5]. Most DFUs are developed in diabetes patients with a concurrence of several risk factors. Among these risk factors, diabetic peripheral neuropathy and peripheral artery disease play important roles in wound healing [5]. The current standard of DFU management includes metabolic control, pressure offloading, restoration of tissue perfusion, antimicrobial treatment, adequate debridement, and topical therapy [4]. Among those measures, topical therapy is one of the most convenient options for DFU patients. Typically, topical therapies are classified into three categories: hydrating, debriding, and antimicrobial dressings [6]. A proper wound dressing not only provides a protective barrier to contamination and external forces but also offers a moist healing environment to reduce the risk of dehydration and necrosis [4]. Despite the various types of topical dressings that have been developed for wound care, there is insufficient evidence in the literature objectively comparing DFU’s healing outcomes with various dressings [7]. Therefore, the choice of the appropriate topical dressing for DFUs remains surgeon or institution dependent.

Ionic silver is well-known for anti-inflammatory and broad-spectrum antimicrobial effects with a low incidence of resistance, which makes it widely used in wound management [8]. The literature has documented that the mechanism of this antimicrobial action is associated with binding to the bacterial wall, blocking enzymatic replication, generating reactive oxygen species and free radicals, and subsequently causing an antiseptic effect [8,9]. As a result of this ability to reduce the bioburden in infected wounds, numerous modern synthetic dressings have been developed to incorporate silver ions within them. Sustained silver-releasing dressings have been shown to significantly reduce the ulcer area and improve the healing rate compared to non-silver dressings in patients with lower limb ulcers [10,11].

Therefore, this study aims to investigate the wound healing efficacy of a new silver-releasing polyurethane foam dressing (Biatain^®^ Ag Non-Adhesive Foam dressing; Coloplast, Humlebaek, Denmark) as compared to that of 1% silver sulfadiazine (SSD) cream in diabetic outpatients with Wagner Grade 1 to 2 DFUs.

## 2. Materials and Methods

### 2.1. Study Design

This study was conducted in a single medical center and approved by the Institutional Review Board and Ethics Committee before initiation of the study (CGMH IRB100-4138A3). In the preselection period, the inclusion criteria for patient enrollment was adult patients diagnosed with type 2 diabetes mellitus undergoing medical management and who presented with a Wagner Grade 1 to 2 DFU with a DFU area of at least 1 cm^2^ according to planimetry, an ankle-brachial index (ABI) in the affected foot of more than 0.7, and a skin perfusion pressure (SPP) in the affected foot of at least 30 mmHg. The exclusion criteria included a known malignancy in the extremities near the target ulceration, an allergy to components of the applied dressings, severe infection such as osteomyelitis, uncontrolled abscess and cellulitis, human immunodeficiency virus infection, uncontrolled medical conditions, and an inability to wear offloading devices. For patients with an ABI in the affected foot less than 0.7 or an SPP less than 30 mmHg angioplasty was completed prior to study entry, thus ensuring that the blood supply of the target DFU was not compromised.

Eligible patients were randomly allocated to a four-week course of treatment in either the treatment or control groups according to computer-generated random numbers (Figure 1). The treatment group received a sterile silver-releasing polyurethane foam dressing (Biatain^®^ Ag Non-Adhesive Foam dressing; Coloplast, Humlebaek, Denmark). For the control group 1% silver sulfadiazine (SSD) cream was used as the primary dressing.

### 2.2. Assessments and Procedures

Five weekly visits were scheduled at week 0, week 1, week 2, week 3, and week 4 for each patient. At the first visit (week 0), the baseline characteristics, past medical history, and information on diabetes-related clinical issue—including duration, medication for diabetes, and levels of hemoglobin A1c (HbA1c)—of each patient were recorded. A clinical evaluation was performed to assess the possible etiology of each DFU wound. An assessment of peripheral vascular circulation was made by measuring SPP on the dorsum of the affected foot and measuring ABI from the dorsalis pedis (DP) and posterior tibial (PT) arteries on both sides. Before clinical evaluation, the DFU wounds were cleaned with sterile sodium chloride solution and mechanically debrided in the outpatient clinic. The microbial culture was collected by a wound swab or tissue biopsy, and antibiotics were prescribed according to the susceptibility of the wound culture report. The wound area was measured with the planimetry method by using transparent acetate tracing sheets,^12^ and digital images of the wounds were taken. The scanned tracing sheets were analyzed with Image J (National Institute of Health, Bethesda, MD, USA) for measurement.

### 2.3. Treatment and Follow-Up Evaluation

In every scheduled visit, all wounds and surrounding tissues were carefully cleaned and irrigated with sterile sodium chloride solution followed by adequate debridement if necessary, ensuring that no visible necrotic tissue was left in the wound. The wound area was measured with the planimetry method before applying the dressings. The dressings used were Biatain^®^ Ag Non-Adhesive Foam dressing, applied at least every two days, in the treatment group and 1% SSD cream, applied once or twice per day, in the control group. Biatain^®^ Ag Non-Adhesive Foam was pretrimmed to an adequate size based on the wound area by the investigator at the outpatient clinic. The frequency of the dressing changes was based on the clinical status of the wound and the amount of exudate. The 1% SSD cream was covered by gauze and tape as a secondary dressing. Step-by-step instructions for self-wound care were provided to the patients and caregivers. The investigator closely assessed the quality of wound care. Offloading devices for plantar ulcers and appropriate shoes were applied to all patients.

### 2.4. Statistical Analysis

Descriptive data were recorded as percentages for categorical variables. The main efficacy parameters were the proportion of the wound healed and the weekly proportion of the wound healed [12]. The equations were showed as below:$$\mathrm{Proportion}\;\mathrm{of}\;\mathrm{wound}\;\mathrm{healed}=\frac{\mathrm{Initial}\;\mathrm{wound}\;\mathrm{area}\;\left({\mathrm{cm}}^{2}\right)-\mathrm{wound}\;\mathrm{area}\;\mathrm{after}\;\mathrm{treatment}\;\left({\mathrm{cm}}^{2}\right)\;}{\mathrm{Initial}\;\mathrm{woud}\;\mathrm{area}\;\left({\mathrm{cm}}^{2}\right)}\;\mathrm{Weekly}\;\mathrm{proportion}\;\mathrm{of}\;\mathrm{wound}\;\mathrm{healed}=\frac{\mathrm{Wound}\;\mathrm{area}\;\mathrm{at}\;\mathrm{week}\;x\;\left({\mathrm{cm}}^{2}\right)-\mathrm{wound}\;\mathrm{area}\;\mathrm{at}\;\mathrm{week}\;x+1\;\left({\mathrm{cm}}^{2}\right)\;}{\mathrm{Woud}\;\mathrm{area}\;\mathrm{at}\;\mathrm{week}\;x\;\left({\mathrm{cm}}^{2}\right)}$$

Continuous variables were expressed as the mean and standard error (SE) or median and interquartile range (IQR), depending on whether the data were normally or non-normally distributed. All analyses were performed with SAS 9.1.3 on an intention-to-treat basis. Student’s t-test or Chi-squared test were used for categorical variables. A *p*-value less than 0.05 was considered significant.

## 3. Results

### 3.1. Trial Population

Of the 66 DFU subjects screened for eligibility 6 patients were excluded from the trial. A total of 60 patients were enrolled in the study and allocated to treatment with either SSD or silver foam. 22 patients (37%) were enrolled in the control group treated with SSD, and 38 patients (63%) were enrolled in the group dressed with Biatain^®^ Ag Non-Adhesive Foam (Figure 1).

### 3.2. General Characteristics of the Participants

The mean age was 66 years in the SSD group and 64 years in the silver foam group. The baseline characteristics were comparable in the two groups without statistical significance (Table 1). Among all the microorganisms, *Enterococcus faecalis* and *Staphylococcus aureus* were the two most commonly isolated from the wound culture of both SSD and silver foam groups (Table 1). No remarkable adverse events, such as allergic reactions, pain, maceration, or pigmentation, were reported.

### 3.3. Wound Healing Efficacy Assessment

The silver foam group had a significantly better wound healing effect than the SSD group, as evidenced by a greater proportion of the wound healed between week 0 to week 4 (76.43 ± 7.41% vs. 27.00 ± 4.95%, *p* < 0.001) (Table 2). The proportion of the wound healed each week was also evaluated. During the first three weeks of treatment, the proportion of the weekly wound healing area was significantly increased in the silver foam group as compared with that in the SSD group (Week 0–1, *p* = 0.002; week 1–2, *p* = 0.043; week 2–3, *p* = 0.048) (Table 2). However, during weeks 3 to 4, the proportion of the weekly wound healing did not show statistical significance (Table 2). Additionally, the proportion of the wound healing area in the silver foam group was significantly higher than in the SSD group throughout the total four weeks of treatment (Figure 2A,B).

### 3.4. Factors Associated with Wound Healing

The subgroup analyses were stratified by clinical predisposing factors such as the size of initial wound area, Wagner classification, level of HbA1c, history of end-stage renal disease (ESRD), and the result of the wound culture. The silver foam facilitated wound closure faster than SSD in the population of HbA1c > 7% (59.94 ± 8.00% vs. 14.21 ± 3.72%, *p* = 0.027) and in patients with positive microbial isolates in their wound culture (60.87 ± 4.06% vs. 37.50 ± 5.89%, *p* = 0.020) (Table 3). In contrast, other predisposing factors, such as the wound area at enrollment and history of ESRD, did not show statistical differences in the proportion of the wound healed between silver foam and SSD groups (Table 3).

## 4. Discussion

The main purpose of this study was to assess the clinical efficacy of the silver-releasing foam dressing, Biatain^®^ Ag Non-Adhesive Foam, in wound management of patients with Wagner Grade 1 to 2 DFUs. The results revealed that the silver foam dressing had a superior effect on wound size reduction to that of SSD cream during a total of four weeks of treatment (Table 2 and Figure 2A). This is comparable to a recent meta-analysis that shows a more than 40% reduction in ulceration area at four weeks in the group managed with the Biatain^®^ Ag foam compared with a non-active dressing [13]. In addition, the proportion of the weekly wound healed in the silver foam treatment group was significantly increased during week 0–1, week 1–2, and week 2–3 of treatment compared with that in the SSD group (Table 2). There were no significant differences in the proportion of the wound healed during the week 3–4 measurements between the two groups (Table 2). These suggest a higher proportion of DFUs healed in the group treated with Biatain Ag^®^ foam dressing during the first three weeks of treatment.

Ionic silver is the main antimicrobial agent in both the dressings we applied in this study: 1% SSD cream and Biatain^®^ Ag Non-Adhesive Foam dressing. The key factors for silver foam to outperform SSD treatment in wound healing are the combination of stable ionic silver release and better exudate control by the dressing fibers [14,15,16]. Compared with Biatain^®^ Ag Non-Adhesive Foam, which enables stable ionic silver release for 7 days after the application of the dressing [13], SSD cream provides a high initial level of silver release but is not able to maintain residual ionic silver activity [17]. Lázaro-Martínez et al. found that the silver dressing significantly decreased a wide spectrum of the bacterial load, including *Staphylococcus aureus*, *Enterobacteriaceae*, and *Pseudomonas aeruginosa*, in diabetic foot ulcers during a 6-week treatment [18]. In our study, there was a higher proportion of DFU wound area healed with the use of the silver foam dressing in the first three weeks of treatment. A possible explanation for this outcome is that early intervention with a silver-releasing dressing decreases the formation of biofilms, thereby reducing the bacterial burden and improving wound healing in the early period of treatment [19].

We also found that the improvement of wound healing in both groups was not sustained after four weeks. This finding might be explained by in vitro and in vivo studies that suggest the cytotoxicity of silver ions toward keratinocytes and fibroblasts leads to delayed re-epithelialization in wound healing during the later treatment period [20].

A large cohort study by Prompers et al. described that the prevalence of infected DFUs was about 60% clinically [21]. Research has suggested that biofilms play a critical role in the etiology of poor-healing DFUs [22]. The presence of a biofilm acts as a protector of bacteria and develops tolerance toward human immune cells and antimicrobial agents, leading to colonization, infection, prolonged inflammation, and impaired wound healing [23,24]. An in vitro study on the antibiofilm effectiveness of silver dressings demonstrates that ionic silver appears to be effective against biofilms by reducing bacterial adhesions and interrupting the extracellular matrix of biofilms [25]. Also, ionic silver is recognized as an antimicrobial agent that kills a broad spectrum of bacteria, including *Staphylococcus aureus, Enterococcus faecalis, Escherichia coli, Pseudomonas aeruginosa, Klebsiella pneumonia*, and even drug-resistant bacteria [25,26]. In our study, the two most common microorganisms were *Enterococcus faecalis* and *Staphylococcus aureus,* which would have been effectively treated by ionic silver [27,28]. This could partly explain why sliver dressings have the positive effect of accelerating wound healing in the early period.

The pathophysiologic mechanism of DFU is associated with a persistent inflammatory phase [5]. The primary factors of this wound healing dysfunction are related to hyperglycemia and peripheral artery disease, which result in impaired leukocyte function, followed by a disturbance of microbial clearance, and increasing risk of infection [23,29,30]. By analyzing the predisposing factors, our study showed that the silver foam significantly reduced the wound size as compared with the SSD treatment in patients with HbA1c > 7% and a positive wound culture. Similar wound healing effects were detected in patients with a large wound area (>10 cm^2^) at enrollment and ESRD history but without a significant difference between the two groups (Table 3). This could relate to the suppression of cellular immunity in diabetic patients, especially the population with a high level of blood glucose, which causes increased susceptibility to infection [21,31].

There are some limitations in this study, including a relatively small number of participants and variable wound conditions. First, the strict criteria of patient selection, especially the exclusion of patients with a suspicion of compromised peripheral blood supply, led to a limited number of eligible patients. In addition, the presence of multiple comorbidities, which results in the fluctuation of general health, is common in diabetic patients and resulted in exclusion from the study. Although the silver dressings are not recommended by current International Working Group on the Diabetic Foot (IWGDF) guidelines due to insufficient evidence [32] it would be interesting to recruit multicenter data to compare other silver dressings, or the same dressing without silver, in order to provide more strong evidence for clinical guidance. Finally, this study included patients with DFU ranging only from Wagner Grade 1 to 2, which is superficial ulcers to deep ulcers, but without osteomyelitis, abscess, or gangrene. Further studies addressing the silver ion dressing on different severity of DFUs are necessary.

## 5. Conclusions

In summary, our study demonstrated that the silver-releasing foam was more effective than traditional silver sulfadiazine cream in terms of wound healing, particularly in the infected DFU and the early period of wound care. Alternative wound dressings are necessary to maintain the rate of wound healing and prevent the negative effects of silver ions on the wound healing rate after three weeks of sliver dressing treatment. However, good blood sugar control is an important factor to enhance diabetic wound healing. Further studies comparing non-silver ion dressings to silver ion-based dressing in the later period of wound treatment would be ideal to assist clinicians with better guidelines for wound dressing and management.

## Figures and Tables

**Figure 1 jcm-10-01495-f001:**
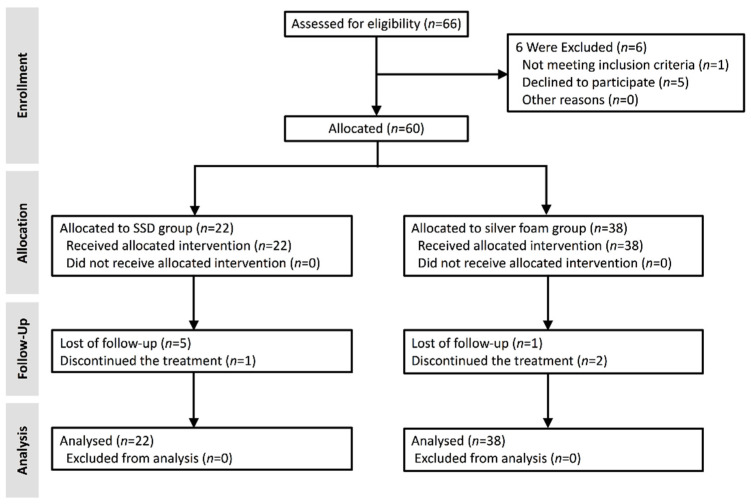
CONSORT flow diagram of the study. Six of the 66 diabetic foot ulcer (DFU) patients were excluded from the trial and 60 patients were included. Of the 22 participants in the 1% silver sulfadiazine (SSD) group who were included in the analysis 1 patient stopped the treatment, and 5 patients were lost to follow-up. Of the 38 participants in the silver foam group, all 38 were included in the analysis, 2 patients discontinued the treatment, and 1 patient was lost to follow-up.

**Figure 2 jcm-10-01495-f002:**
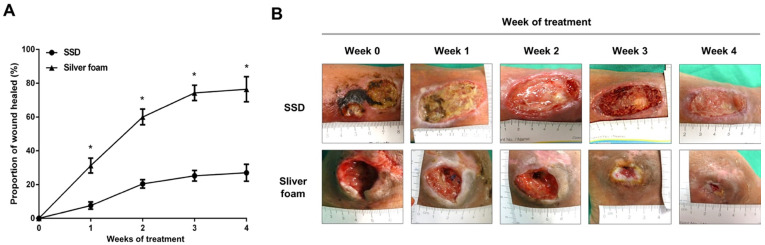
Silver foam dressing facilitated the DFU healing as compared with the SSD dressing. (**A**) The proportion of the wound healed was demonstrated by the percentage of the wound area reduction (week 1, week 2, week 3, and week 4) from the initial wound area at enrollment (week 0). * *p*-value < 0.05. (**B**) The serial follow-up of the patients who received adequate debridement and wound care with the SSD dressing ((**B**), upper panel) and silver foam dressing ((**B**), lower panel) in week 1, week 2, week 3, and week 4 of the treatment.

**Table 1 jcm-10-01495-t001:** Demographic characteristics of the patients between both groups.

Characteristics	SSD	Silver Foam	*p*-Value
Gender (*n*, %)			0.325
Male	15 (68.18%)	21 (55.26%)	
Female	7 (31.82%)	17 (44.74%)	
Age (years)	66 (51–89)	64 (35–88)	0.365
ABI, PT	1.24 (0.72–1.72)	1.15 (0.78–1.9)	0.520
ABI, DP	1.13 (0.74–1.71)	1.08 (0.7–1.9)	0.706
SPP (mmHg)	52.01 (31.9–91.6)	47.20 (32.1–109.7)	0.951
HbA1c (%)	7.4% (5.3–17.7%)	8.1% (5.4–13.1%)	0.064
ESRD			0.093
Yes	8 (36.36%)	6 (15.79%)	
No	14 (63.63%)	32 (84.21%)	
Previous angioplasty in affected limb (%)			0.374
Yes	5 (22.72%)	8 (21.05%)	
No	17 (77.27%)	30 (78.95%)	
Wound area (cm^2^)	6.84 (1.44–19.25)	7.78 (1–30)	0.917
Wagner classification (*n*, %)			0.889
Grade 1	7 (31.82%)	14 (36.84%)	
Grade 2	15 (68.18%)	24 (63.16%)	
Ulcer location (*n*, %)			0.734
Non-plantar	10 (45.45%)	19 (50.00%)	
Plantar	12 (54.54%)	19 (50.00%)	
Microbial isolate (*n*, %)			N/A
*Enterococcus faecalis*	5 (41.67%)	4 (26.67%)	
*Staphylococcus aureus*	3 (25.00%)	4 (26.67%)	
*Proteus mirabilis*	1 (8.33%)	3 (20.00%)	
*Pseudomonas aeruginosa*	1 (8.33%)	2 (13.33%)	
*Escherichia coli*	1 (8.33%)	1 (6.67%)	
*Klebsiella pneumoniae*	1 (8.33%)	1 (6.67%)	

Abbreviation: ABI, ankle brachial index; DP, dorsalis pedis; HbA1c, hemoglobin A1c; PT, posterior tibial; SPP, skin perfusion pressure; SSD, silver sulfadiazine; N/A, not applicable.

**Table 2 jcm-10-01495-t002:** Analysis of wound healing between both groups.

Proportion of Wound Healed (%)	SSD	Silver Foam	*p*-Value
Weekly			
Week 0–1	7.58 ± 1.00	31.24 ± 3.8	* 0.002
Week 1–2	21.03 ± 3.00	50.41 ± 8.92	* 0.043
Week 2–3	21.31 ± 4.78	54.69 ± 6.44	* 0.048
Week 3–4	17.32 ± 4.94	24.92 ± 9.12	0.590
Total			
Week 0–4	27.00 ± 4.95	76.43 ±7.41	* <0.001

* *p*-value < 0.05.

**Table 3 jcm-10-01495-t003:** Analysis of the predisposing factors and wound healing in both groups.

Predisposing Factors	Proportion of Wound Healed (%)		*p*-Value
	SSD	Silver Foam	
Wound area at enrollment			
≥10 cm^2^	32.70 ± 16.20%	52.01 ± 6.70%	0.342
<10 cm^2^	47.95 ± 9.01%	59.96 ± 9.8%	0.630
HbA1c			
>7%	14.21 ± 3.72%	59.94 ± 8.00%	* 0.027
≤7%	47.75 ± 6.56%	55.86 ± 8.50%	0.763
ESRD			
Yes	13.57 ± 5.37%	47.05 ± 5.23%	0.121
No	61.47 ± 7.91%	61.09 ± 6.14%	0.491
Wound culture			
Positive	37.50 ± 5.89%	60.87 ± 4.06%	* 0.020
Negative	49.02 ± 7.90%	74.97 ± 5.25%	0.061

* *p*-value < 0.05.

## Data Availability

The data presented in this study are available on request from the corresponding author. The data are not publicly available due to privacy and ethical concerns.

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
