# Peer review of "The Effects of Silver-Releasing Foam Dressings on Diabetic Foot Ulcer Healing"

_jcm, 2021, doi:10.3390/jcm10071495_

Round 1

Reviewer 1 Report

Your research is very interesting regarding the application of the dressing with positive effects to the closure of diabetic foot ulcers wound, as for the revision, I have found a couple of details to correct, in line 127 and 128 it is written in form wrong "woud" being the correct word "wound".

In the discussion, particularly between the lines 210-213 mentioned that "silver realeasing dressing" decreases the formation of the biofilm, reducing the burden of bacteria, I found an investigation that could be of interest to consolidate his theory, you can find it in the following link "

Lázaro-Martínez JL, Álvaro-Afonso FJ, Sevillano-Fernández D, Molines-Barroso RJ, García-Álvarez Y, García-Morales E. Clinical and Antimicrobial Efficacy of a Silver Foam Dressing With Silicone Adhesive in Diabetic Foot Ulcers With Mild Infection. Int J Low Extrem Wounds. 2019 Sep;18(3):269-278. doi: 10.1177/1534734619866610. Epub 2019 Aug 5. PMID: 31379224.”

Author Response

Dear Reviewer:

         Thank you very much for reviewing our manuscript (jcm-1129323) entitled The Effects of Silver-releasingFoam Dressings on Diabetic Foot Ulcer Healing. We have carefully followed your suggestions to revise this article. The detailed revisions have been highlighted and address the changes point-by-point and to the reviewer’s comments as follows:

Responses to Reviewer’s specific comments:

Reviewer 1

Your research is very interesting regarding the application of the dressing with positive effects to the closure of diabetic foot ulcers wound, as for the revision, I have found a couple of details to correct.

Q1:In line 127 and 128 it is written in form wrong "woud" being the correct word "wound".
Ans. 1:  Thank you for your comments. The typographical errors were corrected in lines 127 and 128.

Q2:In the discussion, particularly between the lines 210-213 mentioned that "silver releasing dressing" decreases the formation of the biofilm, reducing the burden of bacteria, I found an investigation that could be of interest to consolidate his theory, you can find it in the following link "

Lázaro-Martínez JL, Álvaro-Afonso FJ, Sevillano-Fernández D, Molines-Barroso RJ, García-Álvarez Y, García-Morales E. Clinical and Antimicrobial Efficacy of a Silver Foam Dressing With Silicone Adhesive in Diabetic Foot Ulcers With Mild Infection. Int J Low Extrem Wounds. 2019 Sep;18(3):269-278. doi: 10.1177/1534734619866610. Epub 2019 Aug 5. PMID: 31379224.
Ans. 2: As your comments, we have cited this article in refence 18 and discussed in line 211-214.

Reviewer 2 Report

There is no support in the recommendations of the IWGDF guidelines on the use of Argentine sulfadiazine either for accelerating the healing of diabetic foot ulcers or for the management of mild infections. Therefore, it does not make sense to apply this treatment in the control group of the study proposed by the authors. They should have either compared the same dressing without silver to demonstrate the effectiveness of silver or they should have compared it to another silver dressing. For this reason, this study is not attractive or of clinical applicability, beyond the promotion of the product.

Author Response

Dear reviewer:

         Thank you very much for reviewing our manuscript (jcm-1129323) entitled The Effects of Silver-releasingFoam Dressings on Diabetic Foot Ulcer Healing. We have carefully followed the your suggestions to revise this article. The detailed revisions have been highlighted and address the changes point-by-point and to the reviewer’s comments as follows:

Responses to Reviewer’s specific comments:

Q1: There is no support in the recommendations of the IWGDF guidelines on the use of Argentine sulfadiazine either for accelerating the healing of diabetic foot ulcers or for the management of mild infections. Therefore, it does not make sense to apply this treatment in the control group of the study proposed by the authors. They should have either compared the same dressing without silver to demonstrate the effectiveness of silver or they should have compared it to another silver dressing. For this reason, this study is not attractive or of clinical applicability, beyond the promotion of the product.

Ans. 1:As your comment, we understand the IWGDF guidelines do not recommend routinely use silver preparations in treating DFU infection due to insufficient evidence of silver containing treatment in promoting wound healing, except some studies reported that topical silver treatment decreased bacterial load (IWGDF, 2019, recommendation 27 and the rationale). However, numerous studies described sliver dressing have positive effect on wound healing. Therefore, we strived to investigate more evidence about the wound healing efficacy of different silver dressings on DFU. Most of study designs compared silver-releasing dressing and the direct topical silver cream (1% silver sulfadiazine, SSD). Our pilot study demonstrated that the silver foam was effective in the first 3 weeks of the treatment for DFU wound healing, but no apparent effect after the 4th week of the treatment. Further studies are needed to compare the other silver dressing or the same dressing without silver, which are practical issues to reconfirm our preliminary outcome. We have added this in the discussion section in lines 254-258.

We would again like to thank you for the suggestions to improve our manuscript. We hope that the revised manuscript is suitable for publication in Journal of Clinical Medicine.

Sincerely yours,

Yur-Ren Kuo M.D., Ph.D., FACS,

Division of Plastic Surgery, Department of Surgery,

Kaohsiung Medical University Hospital,

100 Tzyou 1st Rd., Kaohsiung 80756, Taiwan.

Phone: +886-7-3121101, ext. 7675, Fax: +886-7-7311482

E-mail: kuoyrren@gmail.com; t1207816@ms22.hinet.net

Reviewer 3 Report

This paper addresses an important problem associated with diabetic foot syndrome. Local treatment should always be preceded by diagnostics and implementation of causal treatment. Authors of this study examined ankle-brachial index and on this basis also qualified patients for angioplasty. I would like to ask if the researchers performed the toe-arm index test in patients, which is dedicated to patients with diabetes. Another question is how many patients underwent angioplasty. And finally, I would like to ask how the patients were assigned to the certain groups. 

Author Response

Dear reviewer:

Thank you very much for reviewing our manuscript(jcm-1129323) entitled The Effects of Silver-releasingFoam Dressings on Diabetic Foot Ulcer Healing.  We have carefully followed the your suggestions to revise this article. The detailed revisions have been highlighted and address the changes point-by-point and to the reviewer's comments as follows:

Responses to Reviewer’s specific comments:

Reviewer 3

This paper addresses an important problem associated with diabetic foot syndrome. Local treatment should always be preceded by diagnostics and implementation of causal treatment. Authors of this study examined ankle-brachial index and on this basis also qualified patients for angioplasty.

Q1:I would like to ask if the researchers performed the toe-arm index test in patients, which is dedicated to patients with diabetes.

Ans. 1: We appreciate your suggestion. We agree that the toe-arm index is a good reference tool for patients with diabetic foot syndrome. However, we did not routinely examine this index because studies have shown skin perfusion pressure (SPP) at dorsal foot or plantar is a good monitoring method to detect the distal foot circulation, and SPP of the foot highly correlated with toe-arm indexand could be substituted for toe-arm index(Pan et al., 2018; Shimazaki et al., 2008). Therefore, we used SPP and ankle-brachial index in diabetic patients to evaluate the condition of peripheral blood perfusion.     

Q2:Another question is how many patients underwent angioplasty.

Ans. 2:As your concern, of the all the 60 participants in this study, 13 patients underwent angioplasty. These 13 patients were allocated into SSD group and silver foam group without statistical significance (SSD vs. silver foam= 22.72% vs. 21.05%,p=0.374). The result was added in Table 1. 

Q3:And finally, I would like to ask how the patients were assigned to the certain groups.

Ans. 3:Thank you for the concern. The patients were allocated into the certain groups by simple randomization with computer-generated random numbers. We have revised this in lines 89-90.  

We would again like to thank you for the suggestions to improve our manuscript. We hope that the revised manuscript is suitable for publication in Journal of Clinical Medicine.

Sincerely yours,

Yur-Ren Kuo M.D., Ph.D., FACS,

Division of Plastic Surgery, Department of Surgery,

Kaohsiung Medical University Hospital,

100 Tzyou 1st Rd., Kaohsiung 80756, Taiwan.

Phone: +886-7-3121101, ext. 7675, Fax: +886-7-7311482

E-mail: kuoyrren@gmail.com; t1207816@ms22.hinet.net

Round 2

Reviewer 2 Report

the use of Argentine sulfadiazine, used for the control group, is not a local treatment protected by the IWGDF consensus guidelines . Therefore, it does not apply to make changes in terms of methodology but in terms of the choice of treatment of the control group, an aspect that is impossible to modify.